# Differential Optimization Federated Incremental Learning Algorithm Based on Blockchain

Xuebin Chen [1,2,3], Changyin Luo [1,2,3,*], Wei Wei [4,5,*], Jingcheng Xu [1,2] and Shufen Zhang [1]

1 College of Science, North China University of Science and Technology, Tangshan 063210, China
2 Hebei Province Key Laboratory of Data Science and Application, North China University of Science and Technology, Tangshan 063210, China
3 Tangshan Key Laboratory of Data Science, North China University of Science and Technology, Tangshan 063210, China
4 School of Computer Science and Engineering, Xi'an University of Technology, Xi'an 710048, China
5 Shaanxi Key Laboratory for Network Computing and Security Technology, Xi'an 710048, China
* Correspondence: lcy20212021@163.com (C.L.); weiwei@xaut.edu.cn (W.W.)

**Abstract:** Federated learning is a hot area of concern in the field of privacy protection. There are local model parameters that are difficult to integrate, poor model timeliness, and local model training security issues. This paper proposes a blockchain-based differential optimization federated incremental learning algorithm, First, we apply differential privacy to the weighted random forest and optimize the parameters in the weighted forest to reduce the impact of adding differential privacy on the accuracy of the local model. Using different ensemble algorithms to integrate the local model parameters can improve the accuracy of the global model. At the same time, the risk of a data leakage caused by gradient update is reduced; then, incremental learning is applied to the framework of federated learning to improve the timeliness of the model; finally, the model parameters in the model training phase are uploaded to the blockchain and synchronized quickly, which reduces the cost of data storage and model parameter transmission. The experimental results show that the accuracy of the stacking ensemble model in each period is above 83.5% and the variance is lower than $10^{-4}$ for training on the public data set. The accuracy of the model has been improved, and the security and privacy of the model have been improved.

**Keywords:** federated learning; blockchain; incremental learning; differential privacy

## 1. Introduction

Google proposed a new privacy protection technology—federated learning—in 2016 [1–3]. Due to its advantages of protecting privacy and local data security, it is widely used in many fields.

Federated learning can be applied to the fields of machine learning and deep learning.

In the field of machine learning, Yang, K. et al. [4] propose an implementation method of vertical federated logistic regression under the central federated learning framework, which can realize the logistic regression in vertical federated learning. However, it is difficult to find a third-party auxiliary party that both partners trust in real life. Therefore, Yang, S.W. et al. [5] proposed a vertical federated logistic regression method under the decentralized federated learning framework. The data between the partners are always confidential, and the transmission channel is also confidential. Liu, Y. et al. [6] proposed a framework based on a central longitudinal federated study method of random forests—federal forest. In the process of modeling, each tree is joint modeling and its structure is stored in each data center server, and the parties have the right to hold it, but each party holds data matching with their characteristics of scattered node information, while unable to obtain useful information from other data-holding parties, to ensure the privacy of data. Finally, the structure of the whole random forest model is scattered and stored, the central

server retains the complete structure information, and the node information is scattered among all data holders. When using the model, the node information stored locally is first used, and then the distributed nodes are coordinated by the central server. This method reduces the communication frequency of each tree during the prediction, which is helpful to communication efficiency. A decentralized horizontal federated learning framework for multi-party GBDT modeling, federated learning based on similarity, is proposed by Li, Q.B. et al. [7] and compensates for communication efficiency but sacrifices a small amount of privacy protection performance. Hartmann V. et al. [8] put forward a method of deploying support vector machines in federated learning, mainly protecting data privacy using feature hash and update block.

In the field of deep learning, Zhu, X.H. et al. [9–11] used a simple CNN to test the existing federated learning framework and the impact of the number of data sets and clients on the federated model. Li, T. et al. [12] proposed FedProx, a federated learning framework for solving statistical heterogeneity, to train LSTM classifiers in the federated data set, which is mainly used for emotion analysis and character prediction.

The training data of federated learning come from different data sources, so the distribution and quantity of training data become the conditions that affect the federated model. If the training data distribution of the data sources is different, it becomes difficult to integrate the local models of multiple parties [13,14]. The logistic regression model was used as the initial global model to train the data of all data sources, and the neural network was used to integrate the local model. However, the performance of the neural network model was non-convex, so it was difficult to achieve the optimal loss function of the model after averaging the parameters. Aiming at this problem, the federated average algorithm (FedAvg) is proposed to integrate the local models of multiple parties with the average value of weights or gradients to obtain the integrated global model [15–18]. However, the gradient depth leak algorithm is proposed for the federated average algorithm, which can restore most of the training data problems according to the gradient update of the local model [19,20]. At the same time, the timeliness of federated models and the security of data and models when training local models are not considered in the above references.

Based on the above problems, this paper puts forward the differential optimization federated incremental learning algorithm based on the blockchain. First, the exponential mechanism and Laplace mechanism are used in the training process of local models to enhance the security of local models and data, but this will result in a loss of accuracy of local models. Second, by optimizing the parameters and weights of decision trees to improve the accuracy of local random forests, we can alleviate the problem of precision loss caused by adding differential privacy technology to local models. Third, using a stacking ensemble algorithm to integrate multiple local model parameters can reduce the risk of information leakage caused by updating local parameters, and further improve the accuracy of the model. Fourth, the initial global model parameters, the local model parameters, and the updated global models are uploaded to the blockchain in each period and quickly synchronized. The consensus algorithm in the blockchain is improved, and a consensus mechanism based on the quality of training parameters (proof of quality, PoQ) [21] is proposed. To improve the efficiency of the blockchain, and from the data storage point of view, storing the model parameters in the blockchain can improve the security and reliability of the model parameters. Experimental results show that the accuracy of the algorithm is higher than that of the federated average algorithm [22], which improves the security of data and model in the training model phase. At the same time, compared with the general federated learning model, the algorithm can automatically upload each time segment and iterated parameters and results to the corresponding data blocks and synchronize quickly, greatly reducing the transmission cost of the model parameters to the blockchain. At the same time, because the blockchain data cannot be tampered with and cannot be deleted, it can protect the model parameters stored on the blockchain.

The main contributions of this paper are as follows:

1. This paper proposes a new federated learning algorithm—differential optimization federated incremental learning algorithm based on blockchain.
2. This method is an experiment on stream data, which verifies the effect of the algorithm on stream data.
3. Considering the risk caused by gradient leakage, the algorithm proposed in this paper applies differential privacy to the algorithm. Gaussian noise is added to the data during model training, and Laplace noise is added to the output of the local model.
4. This experiment is conducted on an unbalanced data set, taking into account the balance between model accuracy and privacy.

The paper is ordered as follows. The algorithm flow and performance analysis are described in the second section. Experimental environment and data set source, data set division to build multi-source stream data, and specific experimental settings are described in Section 3. Finally, conclusions are presented in Section 4.

## 2. Blockchain-Based Differential Optimization Federated Incremental Learning Algorithm

### 2.1. Description of the Algorithm

The blockchain-based differential optimization federated incremental learning algorithm applies differential privacy and ensemble learning to the framework of federated learning. The algorithm includes three stages: model transmission, model training, and model storage. In the model transmission stage, a 512-byte asymmetric encryption algorithm is used to ensure the security of the model transmission process; in the model training stage, differential privacy technology and ensemble algorithms are used to optimize and integrate the parameters while improving the security of the data and the model algorithms to improve the accuracy of the model; in the model storage stage, the blockchain is used to store the parameters of each model in each period, which greatly reduces the cost of data transmission and guarantees the security of the data.

#### 2.1.1. Model Transmission Stage

The algorithm in the model transmission stage is as follows: first, each data source uses the RSA encryption algorithm to generate a 512-byte key pair, and a trusted third party uses the public key to encrypt the initial global model and transmits it to each data source. Each data source uses the private key. Training after decryption ensures the safety of the model type during transmission; each data source uses the private key to encrypt the local model parameters and transmits it to a trusted third party. After the trusted third party uses the public key to decrypt, the ensemble algorithm is used to integrate local model parameters to ensure the safety of local model parameters during transmission, see the purple part in Figure 1.

#### 2.1.2. Model Training Stage

The algorithm in the model training stage is: each data source divides the incremental data into three parts: pre-training set, pre-test set, and test set. The initial global model obtained after decryption with the private key is trained on the pre-training set, and tested on the pre-test set, the score obtained is used as the weight of the base classifier in the initial global model, differential privacy technology is added, the parameters of the model are optimized, a local model that meets privacy protection is established, and the local model is placed on the test set. The training score is used as the local model score of the data source; a trusted third party uses a stacking ensemble algorithm and averaging method to integrate multiple local models to obtain an updated global model for each period, and iterative training is continued, see the green part in Figure 1.

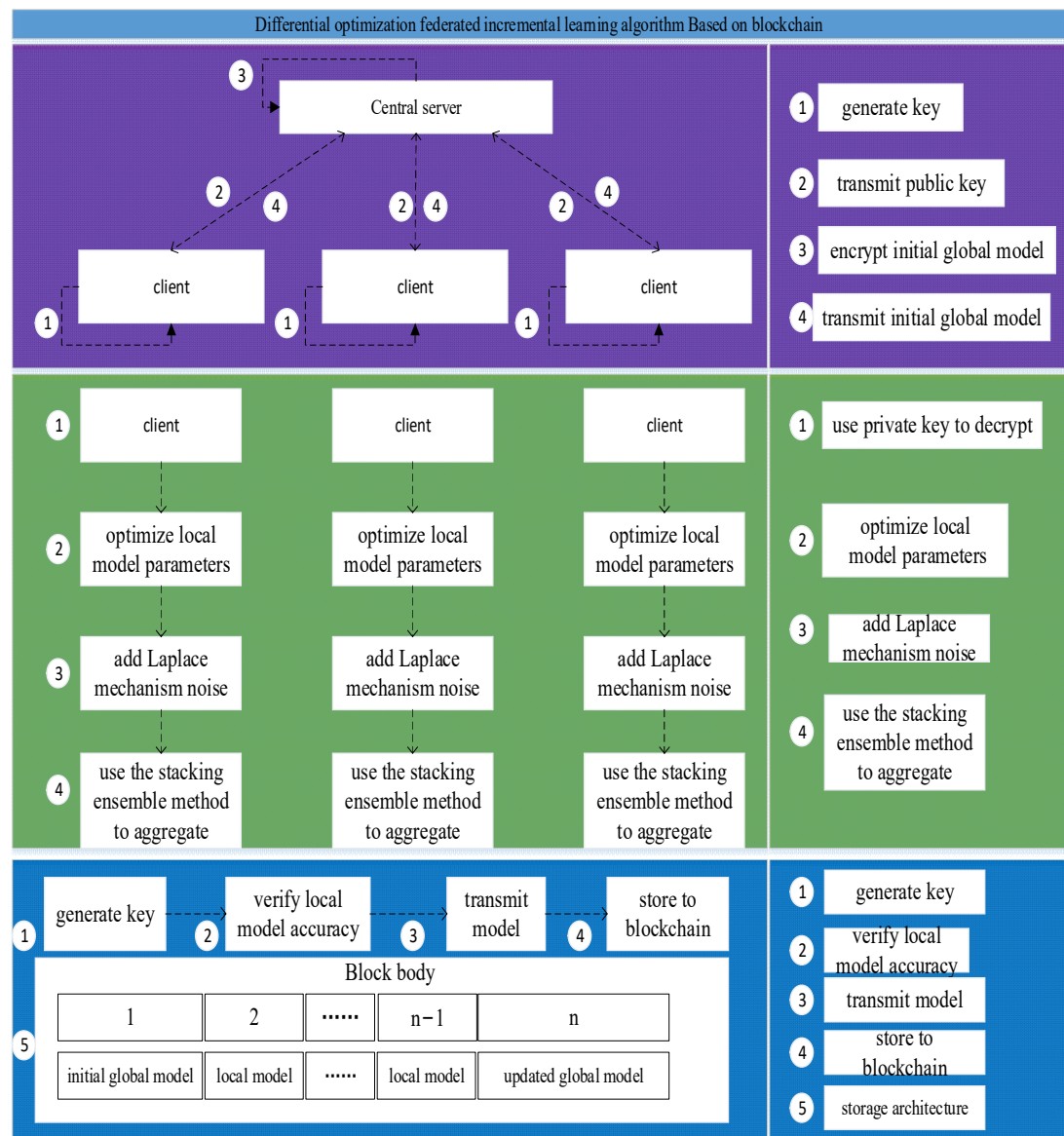

**Figure 1.** Blockchain-based differential optimization joint incremental learning algorithm framework.

### 2.1.3. Model Storage Stage

The algorithm in the model storage stage is: the initial global model parameters in each period are encrypted with a private key and uploaded to the corresponding block. The verification node on the blockchain uses a consensus mechanism based on the proof of training quality. If 2/3 nodes think that the initial global model parameters in this period are the same as the updated global model in the previous period, and they are stored in the corresponding data block 1 in the block generated in this period, for each period, the local model parameters inside are encrypted with a private key and uploaded to the corresponding block. The nodes on the blockchain use a consensus mechanism based on the quality of the training parameters to verify the accuracy of the model. If the accuracy of the local model trained during this period is not up to the lowest accuracy rate determined by the 2/3 nodes, the data source needs to optimize the model parameters to further optimize the local model and improve the accuracy of the local model. Until the accuracy of the local model of the data source meets the requirements, the local model parameters can be changed. Stored in data blocks 2 to $n-1$, the global model updated in each period is encrypted with a private key and uploaded to the corresponding block. Its nodes on the blockchain use a consensus mechanism based on the quality of training parameters for

verification. If 2/3 nodes consider that the updated global model parameters in this period are comparable with the updated global model in the previous period, and the accuracy fluctuates within an acceptable range, then the global model parameters updated in this period can be compared and stored in the data block n corresponding to the block, see the blue part in Figure 1.

The algorithm framework is as follows:

### 2.2. Algorithm Flow

The algorithm flow is as follows:

Model transfer phase

Step 1: Each data source uses the RSA encryption algorithm to generate a 4096-bit public key and private key, and transmits the public key to a trusted third party;

Step 2: A trusted third party uses the public key to encrypt the initial global model and transmits it to each data source;

Step 3: After each data source obtains the local model, use the private key to encrypt the local model and transmit it to a trusted third party. The trusted third party uses the public key to decrypt and integrate multiple local models.

Model training phase

Step 1: Each data source uses the private key to decrypt to obtain the initial global model—random forest;

Step 2: Each data source determines the initial parameters of the algorithm, the number of decision trees: $L$, the number of pre-test samples: $X$, and the pre-pruning parameter: $\varepsilon$;

Step 3: Each data source extracts $L$ training sets $D_t(t = 1, 2, \ldots, L)$ from the incremental data with replacement, and divides the samples in the training set $D_t$ into training data and test data;

Step 4: Each data source selects a certain percentage of data from the training data as pre-training samples, and the remaining samples in the training data are used as pre-test samples;

Step 5: Evenly distribute the privacy protection budget B to each tree $\varepsilon' = \frac{B}{L}$, each layer $\varepsilon'' = \frac{\varepsilon'}{d+1}$, and divide the privacy protection budget of each node into two equal $\varepsilon = \frac{\varepsilon''}{2}$;

Step 6: Randomly select $m$ features from the training sample; if the $m$ features contain $n$ continuous features, assign the privacy protection budget in each node to each continuous feature, and assign a copy to the discrete feature $\varepsilon = \frac{\varepsilon}{n+1}$. For continuous features, use the calculated value from the formula $\frac{\exp(\frac{\varepsilon}{2\Delta q}q(D_c,F))|R_i|}{\sum_i \exp(\frac{\varepsilon}{2\Delta q}q(D_c,F))|R_i|}$ to replace $p_k$ in the formula $Gini(p) = \sum_{k=1}^{K} p_k(1 - p_k) = 1 - \sum_{k=1}^{K} p_k^2$, calculate the corresponding Gini index, and then select the best continuity feature;

Step 7: Compare the Gini index corresponding to the best continuous feature with the Gini index corresponding to each discrete feature, and select the split feature and split point with the smallest Gini index among the randomly selected m features, according to this feature and the best split point, divide the current node into two child nodes, and use Step 6 and Step 7 for each child node;

Step 8: If the node reaches the stopping condition, set the current node as a leaf node, and use the Laplace mechanism to add noise to classify the current node; otherwise, set the current node as a child node, calculate the number of samples of the child node, and use the Laplace mechanism to add noise N = NoisyCount (the number of child node samples), and establish a decision tree that satisfies $\varepsilon$—differential privacy protection;

Step 9: Use the classification accuracy of $L$ decision trees in the pre-test samples as the weight of L decision trees to form a random forest that satisfies $\varepsilon$—differential privacy;

Step 10: Iteratively optimize the parameters in Step 1, select the final optimized parameters, and generate an optimized random forest model that satisfies $\varepsilon$—differential privacy.

Step 11: A trusted third party integrates multiple local models that meet differential privacy using a stacking ensemble algorithm and averaging method to obtain an updated global model.

Model storage stage

Step 1: A trusted third party uses the ECC encryption algorithm to generate a 521-bit key and a private key and transmits the generated private key to each data source while keeping a private key in each period, and the public key is transmitted to the corresponding generated block $i$;

Step 2: Each data source uses the private key to encrypt the initial global model and uploads it to the corresponding generated block $i$. The corresponding block is decrypted with the public key. The nodes on the blockchain adopt a consensus mechanism based on the proof of training quality (PoQ), for verification, if 2/3 nodes believe that the initial global model parameters in this period are the same as the updated global model parameters in the previous period, that is, $w_{h_i} = w_{h_{i-1}}$, then the initial global model parameters in the period are stored in the corresponding data block 1 in the generated block;

Step 3: Each data source uses the private key to encrypt the local model and uploads it to the corresponding block $i$. The corresponding block is decrypted with the public key. The nodes on the blockchain adopt a consensus mechanism based on PoQ for verification. If the accuracy of the local model trained in this period does not reach the minimum accuracy determined by 2/3 nodes, the data source needs to further optimize the local model and improve the accuracy of the local model until the accuracy of the local model meets the requirements, that is, it satisfies the formula $score_{local\_model} \geq \alpha$ ($\alpha$ is the lowest accuracy rate recognized by 2/3 nodes), and the local model parameters can be stored in data blocks 2 to $n-1$;

Step 4: The global model updated in each period is encrypted with a private key and uploaded to the corresponding block, the corresponding block is decrypted with the public key, and the nodes on the blockchain use a consensus mechanism based on the quality of training parameters for verification. If 2/3 nodes think that the updated global model parameters in this period are comparable with the updated global model in the last period, and the accuracy rate fluctuates within an acceptable range, namely $\left| score_{h_i} - score_{h_{i-1}} \right| \leq \beta$ ($\beta$ is the acceptable fluctuation range of 2/3 nodes), then the global model parameters updated in this period can be stored in the data block n corresponding to this block.

The schematic diagram of the storage part is shown in Figure 2:

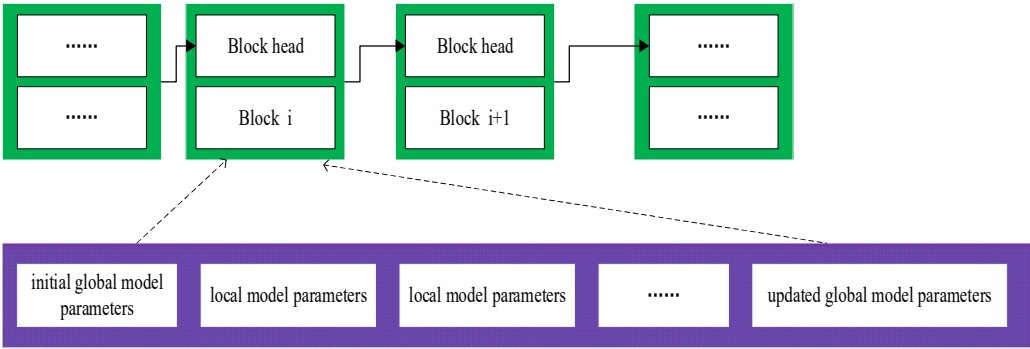

**Figure 2.** Block storage diagram.

### 2.3. Performance Analysis

The federated incremental learning algorithm based on differential optimization based on blockchain proposed in this paper is more accurate than the federated average algorithm, and the security of the model is also improved.

### 2.3.1. Complexity Analysis of the Algorithm

The complexity of the differential optimization federated incremental learning algorithm: the complexity of the RSA encryption algorithm, the complexity of the ECC digital signature algorithm, the complexity of model transmission, the complexity of model update, and the complexity of model storage, that is, the sum of time complexity is $O(((n*\log(n)*d*k)+N^2+w^{k+2}+G^{k+2})*l)$ (where $n$ is the sample, $d$ is the feature, $k$ is the number, $N$ is the complexity of the encryption algorithm, $w$ is the complexity of the initial global model transmission, $G$ is the complexity of the local model transmission, and $l$ is the number of rounds), using the federated learning and stacking ensemble algorithm, will inevitably cause the time complexity of this algorithm to be higher than the federated average algorithm.

### 2.3.2. Security Analysis of the Algorithm

The algorithm uses the federated learning framework and the idea of ensemble learning. In the model training stage, the algorithm is implemented under the framework of federated learning, and the data of each data source are always stored locally, eliminating the risk caused by data transmission, thereby improving the security of the model and data. In the model storage stage, the algorithm uses ECC to generate a key pair by a trusted third party and uses a private key to sign the initial global model parameters, local model parameters, and updated global model parameters in each period and transmit them to the block $i$. The block $i$ is verified using the public key and sequentially stored in the data block, which can ensure the security of the model storage process.

### 2.3.3. Timeliness Analysis of the Algorithm

The algorithm adopts the idea of incremental learning. From the data level, the data generated by each data source in each period are used as the data when the model is trained in that period, to ensure the timeliness of the data level; from the model at that level, the global model is updated in the period and is used as the initial global model for the next period for training, and the local model and the updated global model for the next period can be obtained, thereby ensuring the timeliness of the model level.

### 2.3.4. Privacy Analysis of the Algorithm

The algorithm proposed in this paper first distributes the given privacy protection budget B equally to T trees in the random forest $\varepsilon'=\frac{B}{T}$. Since the samples in each tree are randomly selected again, there will be a certain cross. According to the sequence combination of differential privacy protection, the consumed privacy protection budget is the superposition of the privacy protection budget consumed by each decision tree. The privacy protection budget $\varepsilon$ [23] is equally allocated to the leaf nodes in the tree, namely $\varepsilon=\varepsilon'/2(d+1)$. If the node is a leaf node, the other half of the privacy protection budget is used combined with the Laplace mechanism to add noise to the count value to determine the category of the leaf node; if the node is an intermediate node, the other half of the privacy protection budget is used combined with the index mechanism and Laplace mechanism to select the best split feature and the best split point [14]. The privacy protection budget of each data source is not greater than B, and according to the sequence combination of differential privacy [15,16], this algorithm can meet the requirements of $\varepsilon$—differential privacy protection.

## 3. Analysis of the Experiment

### 3.1. Experimental Parameter Setting

The algorithm is developed and implemented in the Python language and Pycharm integrated software. The experimental hardware environment is: Intel(R) Core i5-4200 MCPU2.50 GHz processor, memory 8 G; the operating system is Windows 10. In terms of experimental data, the data set downloaded from https://www.heywhale.com/mw/dataset/5e61c03ab8dfce002d80191d/file (accessed on 1 June 2022) (Supplementary Material) is

used, and the data set is 15.6 Mb. This data set is the data set in the actual data competition, and has practical significance. Details of the dataset are provided in the supplementary materials.

### 3.2. Analysis of Experimental Data

The data set is randomly divided into 12 parts to represent the data generated by different data sources in different periods, and the result of 20 divisions represents the randomness and rationality of the data. The data set randomly divided 20 times reflects the relationship between the data before and after the division. The randomly divided data set can meet the needs of the same data source and different samples and can meet the rationality of the cross-validation model.

### 3.3. Analysis of Experimental Model

The experiment in this article is divided into three parts: model distribution, model training, and model storage. The first part is to use the random forest as the initial global model and use the public key generated by the RSA encryption algorithm to encrypt and transmit to each data source. Each data source is decrypted with a private key. The initial global model is obtained. The second part: each data source trains the acquired initial global model on incremental data, optimizes the number of trees $L$ and the number of pre-test samples $X$, and the pre-pruning parameter $l$, obtains the period local model, then uses the private key generated by the RSA encryption algorithm to encrypt the local model on each data source and transmit it to a trusted third party. The trusted third party uses the public key to decrypt and integrates the local model with the stacking ensemble algorithm to obtain the most effective updated global model during the period. The third part: a trusted third party uses the ECC digital signature algorithm to generate a key pair, transmits its private key to each data source, and retains a private key, and the public key is transmitted to the corresponding block. Each data source is a trusted third party that uses a private key to sign the initial global model parameters, local model parameters, and updated global model parameters in each period and transmit them to the corresponding block. The block uses the public key to verify and store it in the corresponding data block.

In the model distribution stage, to ensure that each initial global model $H_i$ can be safely transmitted to each data source, the initial global model needs to be encrypted with a 4096-bit public key for transmission.

In the model training phase, each data source uses the private key to decrypt to obtain the initial global model—random forest—uses the random forest to train on each data source, and optimizes the tree $L$ and the number of pre-test samples $X$, and pre-pruning parameters $l$. The optimal parameter model in the $t_1$ period is obtained. Detailed parameter optimization is shown in the Supplementary Material.

The accuracy of the random forest in the $t_1$ period can be expressed as the average of the training accuracy of the random forest on the incremental data generated by each data source in the $t_1$ period, which can ensure the accuracy of each data. Three local models $h_{in}$ are generated each time. To test the performance of the local model, the average value and variance are used to measure. Table 1 shows the performance of the initial global model $H_0$ on each data source.

Among them, $k = 1, 2, 3$ indicates the number of data sources. To see the difference between the variances more clearly, the original variance is magnified by $10^5$. In the following table, $k$ indicates the data source, and the variances are all magnified by $10^5$.

It is obvious from Table 1 that the accuracy of the initial global model $H_0$ trained on the data generated in the $t_1$ period is above 74.5%, and the variance is very small, indicating that the performance of multiple local models is good, and weighting is slightly better than unweighting, and the model without differential privacy is significantly better than the differential privacy protection model with different privacy protection budgets. When the privacy protection budget is 0.25, the accuracy of the model is slightly better than 0.5 and 0.75, indicating privacy while the protection is guaranteed, and the accuracy of the model is also improved.

**Table 1.** The performance of the initial global model $H_0$ in the $t_1$ period.

| Privacy Budget | Weighted or Unweighted | Data Source | Means | Variance |
|---|---|---|---|---|
| 0.25 | weighted | k = 1 | 0.749648 | 3.30 |
| | | k = 2 | 0.74902933 | 4.59 |
| | | k = 3 | 0.745424 | 4.77 |
| | unweighted | k = 1 | 0.75021867 | 5.14 |
| | | k = 2 | 0.74872 | 4.34 |
| | | k = 3 | 0.74356267 | 2.16 |
| 0.5 | weighted | k = 1 | 0.75088 | 6.25 |
| | | k = 2 | 0.74970667 | 3.49 |
| | | k = 3 | 0.74642133 | 4.32 |
| | unweighted | k = 1 | 0.749008 | 4.02 |
| | | k = 2 | 0.750864 | 4.36 |
| | | k = 3 | 0.74667733 | 4.32 |
| 0.75 | weighted | k = 1 | 0.75013867 | 4.84 |
| | | k = 2 | 0.74949867 | 5.38 |
| | | k = 3 | 0.74707733 | 3.34 |
| | unweighted | k = 1 | 0.750416 | 4.15 |
| | | k = 2 | 0.75041067 | 3.45 |
| | | k = 3 | 0.74593067 | 5.68 |
| original | weighted | k = 1 | 0.84373333 | 2.37 |
| | | k = 2 | 0.843712 | 2.08 |
| | | k = 3 | 0.83837333 | 2.05 |
| | unweighted | k = 1 | 0.84249067 | 2.81 |
| | | k = 2 | 0.84322667 | 2.55 |
| | | k = 3 | 0.839104 | 1.87 |

It can be seen from Table 2 that when no differential privacy is added, the accuracy of the weighted model is slightly better than that of the unweighted model. The variance of the weighted model is greater than that of the non-unweighted model, but the variances in Table 2 are all less than $10^{-4}$. For the model with differential privacy, when the privacy protection budget is 0.25, the accuracy of the model is the lowest, followed by the privacy protection budgets of 0.5 and 0.75. It shows that the smaller the privacy protection budget, the lower the availability of the model, but the higher the privacy of the data and the model; compared with the federated average algorithm, the accuracy of the updated global model obtained by the stacking ensemble algorithm is increased by about 5%, and at the same time it is smaller than the variance of the federated average algorithm, indicating that the stability and generalization ability of the model are better than that of the federated average algorithm.

To test the training results of the global model $h_1$ updated in the $t_1$ period, the global model $h_1$ updated in the $t_1$ period is used as the initial global model $H_1$ in the $t_2$ period, and training is performed on the data generated in the $t_2$ period. The initial global model $H_1$ is used to train on each data source, the tree $L$ of the tree, the number of pre-test samples $X$, and the pre-pruning parameter $l$ are optimized, and the model of the optimal parameter in the $t_2$ period is obtained. Detailed parameter optimization is shown in the Supplementary Material.

The accuracy of the random forest in the $t_2$ period can be expressed as the average of the training accuracy of the random forest on the incremental data generated by each data source in the $t_2$ period, which can ensure the accuracy of each data. Three local models are generated each time. To test the performance of the local model, the average value and variance are used to measure. Tables 3 and 4 indicate that the initial global model $H_1$ is the update of each data source using a stacking ensemble during the $t_1$ period. The performance of the updated global model was obtained by the global model and the federated average algorithm in the $t_2$ period.

**Table 2.** A table of changes in the situation of the updated global model $h_1$ over $t_1$ time using different methods.

| Privacy Budget | Weighted or Unweighted | Methods | Means | Variance |
|---|---|---|---|---|
| | | stacking | 0.84379733 | 2.39 |
| 0.25 | weighted | average | 0.805264 | 2.83 |
| | | stacking | 0.84299733 | 2.04 |
| | unweighted | average | 0.80557333 | 4.04 |
| | | stacking | 0.84381333 | 2.22 |
| | weighted | average | 0.805024 | 3.18 |
| 0.5 | | stacking | 0.8444 | 1.66 |
| | unweighted | average | 0.806496 | 3.24 |
| | | stacking | 0.84392533 | 2.25 |
| 0.75 | weighted | average | 0.80664533 | 3.05 |
| | | stacking | 0.84402133 | 3.33 |
| | unweighted | average | 0.80640533 | 1.80 |
| | | stacking | 0.843344 | 2.69 |
| original | weighted | average | 0.843808 | 2.67 |
| | | stacking | 0.84368 | 2.32 |
| | unweighted | average | 0.84405333 | 2.45 |

**Table 3.** The stacking ensemble performance of the initial global model $H_1$ for the $t_2$ period.

| Privacy Budget | Weighted or Unweighted | Data Source | Means | Variance |
|---|---|---|---|---|
| | | k = 1 | 0.841850667 | 2.33 |
| | weighted | k = 2 | 0.842965333 | 1.51 |
| 0.25 | | k = 3 | 0.84672 | 2.03 |
| | | k = 1 | 0.841221333 | 2.04 |
| | unweighted | k = 2 | 0.840698667 | 2.98 |
| | | k = 3 | 0.844698667 | 1.93 |
| | | k = 1 | 0.841344 | 2.05 |
| | weighted | k = 2 | 0.84 | 2.03 |
| | | k = 3 | 0.844874667 | 2.84 |
| 0.5 | | k = 1 | 0.841210667 | 2.86 |
| | unweighted | k = 2 | 0.838474667 | 3.39 |
| | | k = 3 | 0.846522667 | 2.65 |
| | | k = 1 | 0.841978667 | 2.40 |
| | weighted | k = 2 | 0.840890667 | 2.41 |
| | | k = 3 | 0.845226667 | 2.14 |
| 0.75 | | k = 1 | 0.843002667 | 2.14 |
| | unweighted | k = 2 | 0.840112 | 2.75 |
| | | k = 3 | 0.846149333 | 3.47 |
| | | k = 1 | 0.843664 | 2.39 |
| | weighted | k = 2 | 0.839664 | 2.36 |
| original | | k = 3 | 0.84568 | 2.78 |
| | | k = 1 | 0.844496 | 2.86 |
| | unweighted | k = 2 | 0.8408 | 2.81 |
| | | k = 3 | 0.846170667 | 1.97 |

It is obvious from Table 3 that the accuracy of the initial global model $H_1$ trained on the data generated by the $t_2$ period is above 83.5%, and the variance is very small, indicating that the performance of multiple local models is good, the weighted and unweighted accuracy rates are almost equal because the model trained on the data generated in the $t_2$ period have converged, and compared with the $t_1$ period, the accuracy of the differential privacy protection model with different privacy protection budgets is the same as the accuracy of the model without differential privacy.

It is obvious from Table 4 that when the initial global model $H_1$ is a federated average algorithm, the accuracy of the model trained on the data generated in the $t_2$ period is all above 80%, and the variance is very small, indicating multiple local models compared

with the $t_1$ period, and the accuracy of the differential privacy protection model with different privacy protection budgets increased by about 5%, indicating that the model has strong generalization.

**Table 4.** The performance of the initial global model $H_1$ over the $t_2$ period of the federated average algorithm.

| Privacy Budget | Weighted or Unweighted | Data Source | Means | Variance |
|---|---|---|---|---|
| 0.25 | weighted | k = 1 | 0.80474667 | 3.92 |
| | | k = 2 | 0.80414933 | 3.99 |
| | | k = 3 | 0.80739733 | 3.02 |
| | weighted | k = 1 | 0.80486933 | 4.23 |
| | | k = 2 | 0.80140267 | 3.79 |
| | | k = 3 | 0.80955733 | 3.01 |
| 0.5 | weighted | k = 1 | 0.80399467 | 2.95 |
| | | k = 2 | 0.80283733 | 3.16 |
| | | k = 3 | 0.807296 | 4.37 |
| | weighted | k = 1 | 0.80453867 | 4.25 |
| | | k = 2 | 0.80230933 | 3.32 |
| | | k = 3 | 0.806624 | 6.64 |
| 0.75 | weighted | k = 1 | 0.80642667 | 4.62 |
| | | k = 2 | 0.803792 | 3.74 |
| | | k = 3 | 0.80810133 | 3.81 |
| | weighted | k = 1 | 0.80309867 | 3.04 |
| | | k = 2 | 0.80331733 | 3.51 |
| | | k = 3 | 0.807088 | 2.90 |
| original | weighted | k = 1 | 0.84232 | 2.58 |
| | | k = 2 | 0.84066133 | 3.09 |
| | | k = 3 | 0.84746667 | 3.59 |
| | weighted | k = 1 | 0.84256533 | 2.55 |
| | | k = 2 | 0.84032 | 3.07 |
| | | k = 3 | 0.84507733 | 2.18 |

It can be seen from Table 5 that when differential privacy is not added, the accuracy of the weighted model is almost equal to that of the unweighted model. The variance of the weighted model is greater than that of the non-weighted model, but the variances in the table are all less than $10^{-4}$; for the model with differential privacy, when the privacy protection budget is 0.25, the accuracy of the model is the lowest, followed by the privacy protection budgets of 0.5 and 0.75. It shows that the smaller the privacy protection budget, the lower the availability of the model, but the higher the privacy of the data and the model; compared with the federated average algorithm, the accuracy of the updated global model obtained by the stacking ensemble algorithm is increased by about 2%, and at the same time it is smaller than the variance of the federated average algorithm, indicating that the stability and generalization ability of the model are better than that of the federated average algorithm.

To fully test the influence of increasing or reducing data sources on the algorithm, the $t_3$ period is reduced by one data source (two data sources) for training, and the global model $h_2$ updated in the $t_2$ period is used as the initial global model $H_2$ in the $t_3$ period. Training on the data generated in the $t_3$ period is carried out. The initial global model is used to train on each data source, the tree $L$ of the tree, the number of pre-test samples $X$, and the pre-pruning parameter $l$ are optimized, and the model of the optimal parameter in the $t_3$ period is obtained. Detailed parameter optimization is shown in the Supplementary Material.

The accuracy of the random forest in the $t_3$ period can be expressed as the average of the training accuracy of the random forest on the incremental data generated by each data source in the $t_3$ period, which can ensure the accuracy of each data. Two local models $h_{in}$ are generated each time. To test the performance of the local model, the average value and variance are used to measure. Tables 6 and 7 indicate that the initial global model

$H_2$ is the update of each data source using a stacking ensemble during the $t_2$ period. The performance of the updated global model was obtained by the global model and the federated average algorithm in the $t_3$ period.

**Table 5.** Situation change table for updated global $h_2$ models calculated using different methods over $t_2$ time.

| Privacy Budget | Weighted or Unweighted | Methods | Means | Variance |
|---|---|---|---|---|
| 0.25 | weighted | stacking | 0.843301333 | 3.27 |
| | | average | 0.837573333 | 2.17 |
| | unweighted | stacking | 0.844074667 | 2.65 |
| | | average | 0.837616 | 2.99 |
| 0.5 | weighted | stacking | 0.842176 | 2.80 |
| | | average | 0.836944 | 1.68 |
| | unweighted | stacking | 0.84328 | 2.76 |
| | | average | 0.836021333 | 2.19 |
| 0.75 | weighted | stacking | 0.842784 | 4.25 |
| | | average | 0.836458667 | 2.46 |
| | unweighted | stacking | 0.842357333 | 2.49 |
| | | average | 0.836304 | 1.98 |
| original | weighted | stacking | 0.842890667 | 1.97 |
| | | average | 0.842773333 | 2.53 |
| | unweighted | stacking | 0.842906667 | 1.91 |
| | | average | 0.842922667 | 2.07 |

**Table 6.** The stacking ensemble performance of the initial global model $H_2$ for the $t_3$ period.

| Privacy Budget | Weighted or Unweighted | Data Source | Means | Variance |
|---|---|---|---|---|
| 0.25 | weighted | k = 1 | 0.839586667 | 2.42 |
| | | k = 2 | 0.841786667 | 1.36 |
| | unweighted | k = 1 | 0.840706667 | 3.03 |
| | | k = 2 | 0.84352 | 2.22 |
| 0.5 | weighted | k = 1 | 0.8406 | 2.05 |
| | | k = 2 | 0.842213333 | 1.60 |
| | unweighted | k = 1 | 0.840666667 | 2.11 |
| | | k = 2 | 0.844933333 | 2.43 |
| 0.75 | weighted | k = 1 | 0.843333333 | 3.36 |
| | | k = 2 | 0.845373333 | 1.89 |
| | unweighted | k = 1 | 0.841493333 | 2.15 |
| | | k = 2 | 0.843893333 | 1.89 |
| original | weighted | k = 1 | 0.843973333 | 2.48 |
| | | k = 2 | 0.842946667 | 1.47 |
| | unweighted | k = 1 | 0.841082667 | 1.38 |
| | | k = 2 | 0.842 | 1.84 |

It is obvious from Table 6 that the accuracy of the initial global model $H_2$ trained on the data generated in the $t_3$ period is more than 83.5%, and the variance is very small, which shows that the performance of multiple local models is good and has good stability. Compared with the accuracy of the model in the $t_2$ period, the accuracy of the model is improved, which indicates that the initial global model $H_2$ has a strong generalization ability.

It is obvious from Table 7 that the accuracy of the initial global model $H_2$ trained on the data generated in the $t_3$ period is more than 83.5%, and the variance is very small, which shows that the performance of multiple local models is good and has good stability. Compared with the accuracy of the model in the $t_2$ period, the accuracy of the model is improved by more than 3%, which indicates that the initial global model $H_2$ has a strong generalization ability.

**Table 7.** The performance of the initial global model $H_2$ in $t_3$ of the federated average algorithm.

| Privacy Budget | Weighted or Unweighted | Data Source | Means | Variance |
|---|---|---|---|---|
| 0.25 | weighted | k = 1 | 0.83484 | 2.25 |
| | | k = 2 | 0.83756 | 3.30 |
| | unweighted | k = 1 | 0.836253333 | 1.96 |
| | | k = 2 | 0.83624 | 2.14 |
| 0.5 | weighted | k = 1 | 0.834106667 | 2.37 |
| | | k = 2 | 0.838666667 | 2.73 |
| | unweighted | k = 1 | 0.833946667 | 1.96 |
| | | k = 2 | 0.838266667 | 1.41 |
| 0.75 | weighted | k = 1 | 0.83268 | 2.24 |
| | | k = 2 | 0.8396 | 1.45 |
| | unweighted | k = 1 | 0.834413333 | 1.99 |
| | | k = 2 | 0.83792 | 2.23 |
| original | weighted | k = 1 | 0.839413333 | 2.33 |
| | | k = 2 | 0.84612 | 2.62 |
| | unweighted | k = 1 | 0.841586667 | 2.37 |
| | | k = 2 | 0.845813333 | 2.60 |

It can be seen from Table 8 that the accuracy of the weighted model is almost equal to that of the unweighted model without adding differential privacy, and the variance of the weighted model is greater than that of the non-weighted model, but the variance in the table is less than $10^{-4}$; for the model with differential privacy, when the privacy protection budget is 0.25, 0.5, 0.75, the accuracy of the model is almost equal, compared with the federated average algorithm, and the accuracy of the updated global model obtained by the stacking ensemble algorithm is almost equal to that obtained by the federal average algorithm, but the security of the model is improved.

**Table 8.** The $t_3$ time section uses different methods to calculate the change table of the updated global model $h_3$.

| Privacy Budget | Weighted or Unweighted | Methods | Means | Variance |
|---|---|---|---|---|
| 0.25 | weighted | stacking | 0.8388533 | 2.94 |
| | | average | 0.83928 | 2.32 |
| | unweighted | stacking | 0.8416533 | 2.00 |
| | | average | 0.83896 | 3.33 |
| 0.5 | weighted | stacking | 0.84148 | 1.18 |
| | | average | 0.83896 | 1.64 |
| | unweighted | stacking | 0.83972 | 2.54 |
| | | average | 0.8382667 | 2.79 |
| 0.75 | weighted | stacking | 0.8386933 | 4.18 |
| | | average | 0.8398267 | 1.88 |
| | unweighted | stacking | 0.8400133 | 3.29 |
| | | average | 0.8380933 | 1.79 |
| original | weighted | stacking | 0.8411867 | 2.35 |
| | | average | 0.8392667 | 2.97 |
| | unweighted | stacking | 0.839184 | 2.90 |
| | | average | 0.8390667 | 2.88 |

To fully test the influence of increasing or reducing data sources on the algorithm, one data source (four data sources) was added for training in the $t_4$ period, and the global model $h_3$ updated during the $t_3$ period was used as the initial global model $H_3$ in the $t_4$ period. Training on the data generated in the $t_4$ period is carried out. The initial global model $H_3$ is used to train on each data source, and the number $L$ of the tree, the number of pre-test samples $X$, and the pre-pruning parameter $l$ are optimized, to obtain the model of

the optimal parameter in the $t_4$ period. Detailed parameter optimization is shown in the Supplementary Material.

The accuracy of random forest in the $t_4$ period can be expressed as the average value of training accuracy of 20 iterations on incremental data generated by each data source, which can ensure the accuracy of each data and, at the same time, the four local models $h_{in}$ were generated each time. To test the performance of local models, the average value and variance were used to measure. Tables 9 and 10 show the performance of the initial global model $H_3$ in the $t_4$ period of the updated global model obtained by the stacking ensemble algorithm and the federated average algorithm for each data source in the $t_3$ period.

**Table 9.** The stacking ensemble performance of the initial global model $H_3$ for the $t_4$ period.

| Privacy Budget | Weighted or Unweighted | Data Source | Means | Variance |
|---|---|---|---|---|
| 0.25 | weighted | k = 1 | 0.844331822 | 6.71 |
| | | k = 2 | 0.846293333 | 1.17 |
| | | k = 3 | 0.838826667 | 2.99 |
| | | k = 4 | 0.8424 | 2.15 |
| | unweighted | k = 1 | 0.843798346 | 3.76 |
| | | k = 2 | 0.841333333 | 1.28 |
| | | k = 3 | 0.839253333 | 6.41 |
| | | k = 4 | 0.839306667 | 2.18 |
| 0.5 | weighted | k = 1 | 0.843478261 | 6.84 |
| | | k = 2 | 0.840426667 | 5.54 |
| | | k = 3 | 0.839306667 | 2.03 |
| | | k = 4 | 0.840746667 | 2.98 |
| | unweighted | k = 1 | 0.84502534 | 1.09 |
| | | k = 2 | 0.84416 | 1.32 |
| | | k = 3 | 0.842506667 | 1.01 |
| | | k = 4 | 0.842506667 | 3.13 |
| 0.75 | weighted | k = 1 | 0.841291011 | 1.73 |
| | | k = 2 | 0.841813333 | 5.64 |
| | | k = 3 | 0.84528 | 1.20 |
| | | k = 4 | 0.836266667 | 5.24 |
| | unweighted | k = 1 | 0.842197919 | 1.54 |
| | | k = 2 | 0.84224 | 1.58 |
| | | k = 3 | 0.84592 | 1.88 |
| | | k = 4 | 0.842666667 | 3.52 |
| original | weighted | k = 1 | 0.842998133 | 1.76 |
| | | k = 2 | 0.844053334 | 4.70 |
| | | k = 3 | 0.840746667 | 3.48 |
| | | k = 4 | 0.842133333 | 2.37 |
| | unweighted | k = 1 | 0.844118432 | 1.50 |
| | | k = 2 | 0.846186667 | 6.41 |
| | | k = 3 | 0.839146667 | 5.61 |
| | | k = 4 | 0.837813333 | 1.20 |

From Table 9, it can be seen that the initial global model is the updated global model of the stacking ensemble. The accuracy rate of most of the models trained on the data generated in the $t_4$ period is more than 84%, and the variance is very small, indicating that the performance of several local models is good and has good stability.

From Table 10, it can be clearly seen that the initial global model $H_3$ is the updated global model of the federated average algorithm, and the accuracy rate of most of the models trained on the data generated in the $t_4$ period is more than 84%, and the variance is very small, indicating that the performance of several local models is good and has good stability.

It is obvious from Table 8 that when the initial global model $H_3$ is a federated average algorithm, the accuracy of the model trained on the data generated in the $t_4$ period is all above 80%, and the variance is very small, indicating multiple local models compared with

the period $t_3$, and the accuracy of the differential privacy protection model with different privacy protection budgets increased by about 5%, indicating that the model has strong generalization.

**Table 10.** The performance table of the initial global $H_3$ model for the federated average algorithm in the $t_4$ period.

| Privacy Budget | Weighted or Unweighted | Data Source | Means | Variance |
|---|---|---|---|---|
| 0.25 | weighted | k = 1 | 0.844118432 | 1.83 |
| | | k = 2 | 0.843706667 | 1.57 |
| | | k = 3 | 0.841866667 | 1.88 |
| | | k = 4 | 0.841066667 | 2.55 |
| | unweighted | k = 1 | 0.842331288 | 2.46 |
| | | k = 2 | 0.843653333 | 2.98 |
| | | k = 3 | 0.843813333 | 2.11 |
| | | k = 4 | 0.842053333 | 1.51 |
| 0.5 | weighted | k = 1 | 0.843158176 | 1.28 |
| | | k = 2 | 0.841226667 | 4.22 |
| | | k = 3 | 0.841386667 | 3.76 |
| | | k = 4 | 0.839466667 | 1.95 |
| | unweighted | k = 1 | 0.842864764 | 3.36 |
| | | k = 2 | 0.841813333 | 3.37 |
| | | k = 3 | 0.84312 | 1.26 |
| | | k = 4 | 0.8376 | 1.85 |
| 0.75 | weighted | k = 1 | 0.840944252 | 1.80 |
| | | k = 2 | 0.841973333 | 2.96 |
| | | k = 3 | 0.84456 | 1.86 |
| | | k = 4 | 0.8408 | 1.31 |
| | unweighted | k = 1 | 0.840917578 | 3.03 |
| | | k = 2 | 0.839626667 | 7.41 |
| | | k = 3 | 0.843013333 | 1.03 |
| | | k = 4 | 0.83864 | 2.19 |
| original | weighted | k = 1 | 0.842731395 | 2.79 |
| | | k = 2 | 0.8412 | 2.11 |
| | | k = 3 | 0.84312 | 2.79 |
| | | k = 4 | 0.83856 | 2.93 |
| | unweighted | k = 1 | 0.84001067 | 3.06 |
| | | k = 2 | 0.844026667 | 3.04 |
| | | k = 3 | 0.843466667 | 1.00 |
| | | k = 4 | 0.83712 | 2.22 |

It can be seen from Table 11 that the accuracy of the weighted model is greater than that of the unweighted model, and the variance of the weighted model is greater than that of the non-weighted model, but the variances in the table are less than $10^{-4}$, which meets the experimental requirements. For the model with differential privacy, the accuracy of the model is almost equal when the privacy protection budget is 0.25, 0.5, and 0.75. Compared with the federated average algorithm, the accuracy of the updated global model obtained by the stacking ensemble algorithm is almost the same as that obtained by the federated averaging algorithm, but the security of the model is improved.

In the model storage phase, the trusted third party uses the ECC encryption algorithm to generate the key pair with a length of 512. The public key is broadcast to the corresponding block, and the private key is transmitted to each data source separately, and at the same time, one reservation is maintained.

The Raybaas platform can quickly and efficiently build blockchain-based services and applications. The hardware devices have an Intel(R) Core i5-4200 m CPU 2.50 GHz processor. The underlying blockchain is deployed based on a CentOS 7.6 operating system. The stored data include initial global model parameters, local model parameters, and updated global model parameters in the $t_1$ period.

**Table 11.** The change table of updated global model $h_4$ was calculated by different methods in $t_4$ period.

| Privacy Budget | Weighted or Unweighted | Methods | Means | Variance |
|---|---|---|---|---|
| 0.25 | weighted | stacking | 0.83850667 | 1.12 |
| | | average | 0.83501333 | 1.71 |
| | unweighted | stacking | 0.83770667 | 1.37 |
| | | average | 0.83776000 | 8.38 |
| 0.5 | weighted | stacking | 0.83930667 | 7.70 |
| | | average | 0.83933333 | 2.00 |
| | unweighted | stacking | 0.83450667 | 8.75 |
| | | average | 0.83914667 | 2.17 |
| 0.75 | weighted | stacking | 0.83824000 | 5.93 |
| | | average | 0.83981333 | 1.92 |
| | unweighted | stacking | 0.84314667 | 2.23 |
| | | average | 0.83760000 | 3.51 |
| original | weighted | stacking | 0.83930667 | 7.70 |
| | | average | 0.83725333 | 1.02 |
| | unweighted | stacking | 0.83829333 | 1.08 |
| | | average | 0.83946667 | 2.02 |

For storing the initial global model parameters, the trusted third party encrypts the initial global model in the $t_i$ period by using the private key retained in the period and transfers it to the block $i$. The verification nodes in the blockchain are decrypted using the corresponding public key and verified by the consensus mechanism based on the training parameter quality. If 2/3 verification nodes consider that the initial global model parameters in the period are the same as the updated global model parameters in the previous period, namely, the formula $w_{H_i} = w_{h_{i-1}}$, then the initial global model parameters $w_{H_i}$ in the period will be stored in the corresponding data block 1 in the generated block $i$.

For storing local model parameters, the private key within the $t_i$ time of each data source encrypts the local model in the period and uploads it to the corresponding block $i$. The verification nodes in the blockchain are encrypted using the corresponding public key and the consensus mechanism based on the training parameter quality is used for verification. The verification node in the blockchain uses the corresponding public key to decrypt, and uses a consensus mechanism based on the quality of training parameters for verification. If the accuracy rate of the local model trained in this period cannot reach the minimum accuracy rate determined by the node, the data source needs to further optimize the local model to improve the accuracy of the local model until the accuracy of the local model of the data source meets the requirements, that is, the formula $score_{local\_model} \geq \alpha$ ($\alpha$ is the minimum accuracy rate determined by 2/3 nodes) can be met, and the local model parameters can be stored in data block 2 to $n-1$.

For the updated global model parameters, the trusted third party encrypts the updated global model in the $t_i$ period by using the private key retained in the period and transfers it to the block $i$. The verification nodes in the blockchain are decrypted using the corresponding public key and verified by the consensus mechanism based on the training parameter quality. If 2/3 nodes believe that the updated global model parameters in this period are comparable with the updated global model in the previous period, the accuracy fluctuates within an acceptable range, that is, $\left| score_{h_i} - score_{h_{i-1}} \right| \leq \beta$ ($\beta$ is the acceptable fluctuation range of 2/3 nodes), then the global model parameters updated in the period can be stored in the data block $n$ corresponding to the block.

*3.4. Summary of the Experiment*

The algorithm distributes the differentially weighted optimization random forest to each data source and performs training, and uses the stacking ensemble algorithm to integrate multiple local models. The updated global model $h_1$ has an accuracy of

84.3797333%, 84.3813333%, and 84.3925333% on the incremental data in the $t_1$ period. As the data source of the $t_2$ period is consistent with that of the $t_1$ period, the updated global model is distributed to the data source as the initial global model of the $t_2$ period and trained, and the accuracy of using the stacking ensemble is 84.3301333%, 84.2176%, and 84.2784%, respectively, indicating that the updated global model has strong generalization. To reduce one data source in the $t_3$ period, the updated global model $h_2$ was distributed to the data source and trained as the initial global model in the $t_3$ period, and the accuracy of using the stacking ensemble is 83.8853333%, 84.148%, and 83.8693333%. To add one data source in the $t_4$ period, the updated global model $h_3$ is distributed to the data source and trained as the initial global model in the $t_4$ period, and the accuracy of using the stacking ensemble is 83.8506667%, 83.9306667%, and 83.824%. Compared with the federated average algorithm, the accuracy of the differential weighted optimization random forest is increased by up to 5%, and the average period is increased by 1%. At the same time, the security of the model and data during the training process has been greatly improved.

The comparative experiment in this paper is to compare the stacking integration algorithm with the benchmark algorithm FedAvg algorithm in federated learning under different privacy protection budgets. Under the influence of different privacy protection budgets and weights, the experimental results of the federal average algorithm and stack integral algorithm in stream data are shown in the tables.

## 4. Conclusions

This paper proposes a differential optimization federated incremental learning algorithm based on blockchain. Applying differential privacy and incremental learning to the framework of federated learning can enhance the security and timeliness of data and models. The accuracy of the model is affected by the differential privacy adding model. To mitigate the influence of adding differential privacy to the model, the model is weighted. The initial global model parameters, local model parameters, and updated global models of each period are uploaded to the blockchain. The verification nodes in the blockchain are validated by a consensus mechanism based on training parameter quality. The required parameters are stored in the corresponding data blocks according to rules and quickly synchronized, reducing the transport cost data and at the same time guaranteeing the safety of the model parameters. When optimizing model parameters, this paper adopts the idea of a set, that is, the method of selecting the best among the best to select the optimal parameters. However, in the case of high dimensions, the relatively excellent parameters are obtained, which cannot achieve the effect of optimal parameters. Later, optimization algorithms will be used to optimize. In the following work, we will try to apply this algorithm to other privacy protection technologies to further improve the security of data and models based on ensuring model accuracy.

**Supplementary Materials:** The data set of this experiment comes from https://www.heywhale.com/mw/dataset/5e61c03ab8dfce002d80191d/file (accessed on 1 June 2022). There are 200,000 samples in this data set, of about 15.6 Mb, where: caseid represents the case number, which has no practical significance; $Q_1$ represents the information of the first question. The information is encoded into numbers, and the size of the numbers does not represent the real relationship. $Q_k$ represents the information of the $k$-th question. There are a total of 36 questions. Evaluation represents the final audit result; 0 means the claim is granted and 1 means the claim is not approved.

**Author Contributions:** Conceptualization, X.C. and C.L.; methodology, W.W.; software, C.L.; validation, S.Z. and W.W.; formal analysis, J.X.; investigation, J.X.; resources, J.X.; data curation, S.Z.; writing—original draft preparation, C.L.; writing—review and editing, C.L.; visualization, S.Z.; supervision, W.W.; project administration, X.C.; funding acquisition, X.C and W.W. All authors have read and agreed to the published version of the manuscript.

**Funding:** This research was funded by the National Natural Science Foundation of China grant number (U20A20179), and by the Science Foundation of Shanxi Province of China grant number (2021JM-344) and Open Fund for Chongqing Key Laboratory of Computational Intelligence grant number (NO.2020FF02).

**Data Availability Statement:** The data set of this experiment comes from https://www.heywhale.com/mw/dataset/5e61c03ab8dfce002d80191d/file (accessed on 1 June 2022).

**Acknowledgments:** We thank the anonymous reviewers for their valuable comments and suggestions which helped us to improve the content and presentation of this paper. The work was supported by the National Natural Science Foundation of China (U20A20179), and by the Science Foundation of Shanxi Province of China (2021JM-344) and Open Fund for Chongqing Key Laboratory of Computational Intelligence (NO.2020FF02).

**Conflicts of Interest:** The authors declare no conflict of interest.

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
