# Peer review of "Differential Optimization Federated Incremental Learning Algorithm Based on Blockchain"

_electronics, doi:10.3390/electronics11223814_

Round 1

Reviewer 1 Report

The overall contribution is good. I suggest a major revision. The following are my observations: Strengths: 1. This is an important research topic. 2. The structure of the paper is comprehensive. Weaknesses: 1. In the proposed scheme flow of activities is not clear. There should be some flow chart or activity diagram, which can explain the working of proposed scheme. It should have a clear flow of all processes i.e., which process executes first, then the next process and so on. 2. Background and related work: Generally, the section provides many references, but it remains unclear which problems are being solved in the cited literature, where the prior art falls short of the authors expectations and requirements for a secure DL. 3. The novelty and contribution of this paper are limited, and the disadvantages of the proposal were not clearly discussed while the proposed framework provided more features compared with other approaches. 4. Many abbreviations in the paper are referenced only once or twice, the authors are encouraged to reduce their use where possible to aid in legibility. 5. Mathematical analysis of the proposed scheme is weak. 6. There are lot of works that have used DL, FL and blockchain for security and privacy. Authors should include these works in related studies by including a table mentioning the pros and cons of these works. A comparison is missing in this work. For instance authors can use articles such as: PEFL: Deep Privacy-Encoding-Based Federated Learning Framework for Smart Agriculture; A Blockchain-Orchestrated Deep Learning Approach for Secure Data Transmission in IoT-Enabled Healthcare System; Blockchain and Deep Learning for Secure Communication in Digital Twin Empowered Industrial IoT Network; Blockchain and deep learning empowered secure data sharing framework for softwarized uavs

7. There are many more. Authors should do a through literature survey and include all such papers in related studies. 8. Authors should proofread entire manuscript for grammatical mistakes.

Author Response

P2001174 Modification Description

Dear manuscript reviewer/editorial department teacher:

Thank you very much for your review comments on my paper "Differential optimization federated incremental learning algorithm based on blockchain". Each of your review comments has brought important help to the modification of this paper and future research work.

The text of the manuscript has been modified according to the comments of the reviewer. The following modification instructions (in red text) are made for the comments of the reviewer:

Reviewer # 1:

  1. In the proposed scheme flow of activities is not clear. There should be some flow chart or activity diagram, which can explain the working of proposed scheme. It should have a clear flow of all processes i.e., which process executes first, then the next process and so on.

For this opinion, it has been modified as follows:

The algorithm framework is as follows:

Fig.1 Blockchain based differential optimization joint incremental learning algorithm framework

The schematic diagram of the storage part is as follows:

Fig.2  Block Storage Diagram

  1. Background and related work: Generally, the section provides many references, but it remains unclear which problems are being solved in the cited literature, where the prior art falls short of the authors expectations and requirements for a secure DL.

For this opinion, the contribution of this article has been added at the end of the introduction part of the article.

The main contributions of this paper are as follows:

  1. This paper proposes a new federated learning algorithm - Differential optimization federated incremental learning algorithm based on blockchain.
  2. This method is an experiment on stream data, which verifies the effect of the algorithm on stream data.
  3. Considering the risk caused by gradient leakage, the algorithm proposed in this paper applies differential privacy to the algorithm. Gaussian noise is added to the data during model training, and Laplace noise is added to the output of the local model.
  4. This experiment is conducted on an unbalanced dataset, taking into account the balance between model accuracy and privacy.

The paper is ordered as follows. The algorithm flow and performance analysis are described in the second section.Experimental environment and data set source, data set division to build multi-source stream data and specific experimental settings are described in Section 3. Finally, conclusions are presented in Section 4.

  1. The novelty and contribution of this paper are limited, and the disadvantages of the proposal were not clearly discussed while the proposed framework provided more features compared with other approaches.

In view of this opinion, this paper adds the shortcomings of this article in the conclusion part of the article.

When optimizing model parameters, this paper adopts the idea of set, that is, the method of selecting the best among the best to select the optimal parameters. However, in the case of high dimensions, the relatively excellent parameters are obtained, which cannot achieve the effect of optimal parameters. Later, optimization algorithms will be used to optimize.

  1. Many abbreviations in the paper are referenced only once or twice, the authors are encouraged to reduce their use where possible to aid in legibility.

In response to this opinion, the sources of some abbreviations have been cited and their use has been reduced.

  1. Mathematical analysis of the proposed scheme is weak.

In view of this opinion, the formula used in this paper has been proved to be effective by experiment.

  1. There are lot of works that have used DL, FL and blockchain for security and privacy. Authors should include these works in related studies by including a table mentioning the pros and cons of these works. A comparison is missing in this work. For instance authors can use articles such as: PEFL: Deep Privacy-Encoding-Based Federated Learning Framework for Smart Agriculture; A Blockchain-Orchestrated Deep Learning Approach for Secure Data Transmission in IoT-Enabled Healthcare System; Blockchain and Deep Learning for Secure Communication in Digital Twin Empowered Industrial IoT Network; Blockchain and deep learning empowered secure data sharing framework for softwarized uavs

    For this opinion, it has been revised in the references of the article.

  1. There are many more. Authors should do a through literature survey and include all such papers in related studies.

In view of this opinion, it has been revised in the article's references and related research.

  1. Authors should proofread entire manuscript for grammatical mistakes.

For this opinion, the syntax of the full text has been modified.

Reviewer 2 Report

Dear Authors, the work seems promising, but the overall presentation of the topic is slightly disorganized. The uniqueness or novelty of the work needs to be highlighted clearly in a separate section. 

I can see that you have used the default Proof of Quality which is OK, but have you tried other consensus mechanism?

Please add a comparison table showing a comparative study to other closely related work.

I tried to access the dataset but the link given points to a registration page in Chinese, do you have any other publicly available link from where the dataset can be inspected without registration ?

Author Response

P2001174 Modification Description

Dear manuscript reviewer/editorial department teacher:

Thank you very much for your review comments on my paper "Differential optimization federated incremental learning algorithm based on blockchain". Each of your review comments has brought important help to the modification of this paper and future research work.

The text of the manuscript has been modified according to the comments of the reviewer. The following modification instructions (in red text) are made for the comments of the reviewer:

Reviewer # 2:

  1. The uniqueness or novelty of the work needs to be highlighted clearly in a separate section.

For this opinion, the contribution of this article has been added at the end of the introduction part of the article.

The main contributions of this paper are as follows:

  1. This paper proposes a new federated learning algorithm - Differential optimization federated incremental learning algorithm based on blockchain.
  2. This method is an experiment on stream data, which verifies the effect of the algorithm on stream data.
  3. Considering the risk caused by gradient leakage, the algorithm proposed in this paper applies differential privacy to the algorithm. Gaussian noise is added to the data during model training, and Laplace noise is added to the output of the local model.
  4. This experiment is conducted on an unbalanced dataset, taking into account the balance between model accuracy and privacy.

The paper is ordered as follows. The algorithm flow and performance analysis are described in the second section.Experimental environment and data set source, data set division to build multi-source stream data and specific experimental settings are described in Section 3. Finally, conclusions are presented in Section 4.

  1. I can see that you have used the default Proof of Quality which is OK, but have you tried other consensus mechanism?

In response to this opinion, the formula mechanism of basic data quality used in this paper aims to comprehensively consider the trade-off between multi-source data volume and data quality, and is simplified on the basis of literature [19]. Its theory has been fully verified in literature [19].

  1. Please add a comparison table showing a comparative study to other closely related work.

For this opinion, experimental comparison has been added to the experimental summary section 3.4 of the article. The details are as follows:

The comparative experiment in this paper is to compare the stacking integration algorithm with the benchmark algorithm FedAvg algorithm in federated learning under different privacy protection budgets. Under the influence of different privacy protection budgets and weights, the experimental results of federal average algorithm and stack integral algorithm in stream data are shown in the table.

  1. I tried to access the dataset but the link given points to a registration page in Chinese, do you have any other publicly available link from where the dataset can be inspected without registration ?

    For this opinion, you can see the details of the training set when you enter the data set link in this article. The data set of the article has been uploaded as an auxiliary.

Round 2

Reviewer 1 Report

Accept in current form.

Author Response

P2001174 Modification Description

Dear manuscript reviewer/editorial department teacher:

Thank you very much for your review comments on my paper "Differential optimization federated incremental learning algorithm based on blockchain". Each of your review comments has brought important help to the modification of this paper and future research work.

The text of the manuscript has been modified according to the comments of the reviewer. The following modification instructions (in red text) are made for the comments of the reviewer:

Reviewer :

  1. The paper needs proof-reading.

In response to this opinion, the full text has been proofread and revised.

  1. None of the figures are referred in the text. They should be mentioned in the respective paragraphs.

In response to this opinion, modifications have been made as follows:

2.1.1 Model transmission stage

The algorithm in the model transmission stage is as follows: first, each data source uses the RSA encryption algorithm to generate a 512-byte key pair, and a trusted third party uses the public key to encrypt the initial global model and transmits it to each data source. each data source uses the private key. training after decryption ensures the safety of the model type during transmission; each data source uses the private key to encrypt the local model parameters and transmits it to a trusted third party. After the trusted third party uses the public key to decrypt, the ensemble algorithm is used to integrate local model parameters to ensure the safety of local model parameters during transmission,see the purple part in Fig 1.

2.1.2 Model training stage

The algorithm in the model training stage is: each data source divides the incremental data into three parts: pre-training set, pre-test set, and test set. The initial global model obtained after decryption with the private key is trained on the pre-training set, and Tested on the pre-test set, use the score obtained as the weight of the base classifier in the initial global model, add differential privacy technology, and optimize the parameters of the model, establish a local model that meets privacy protection, and place the local model on the test set The training score is used as the local model score of the data source; a trusted third party uses stacking ensemble algorithm and averaging method to integrate multiple local models to obtain an updated global model for each period, and iterative training is continued,see the green part in Fig 1.

2.1.3 Model storage stage

The algorithm in the model storage stage is: the initial global model parameters in each period are encrypted with a private key and uploaded to the corresponding block. The verification node on the blockchain uses a consensus mechanism based on the proof of training quality. If  nodes think that the initial global model parameters in this period are the same as the updated global model in the previous period, and they are stored in the corresponding data block 1 in the block generated in this period; for each period, the local model parameters inside are encrypted with a private key and uploaded to the corresponding block. the nodes on the blockchain use a consensus mechanism based on the quality of the training parameters to verify the accuracy of the model. If the accuracy of the local model trained during this period is not up to the lowest accuracy rate determined by the  node, the data source needs to optimize the model parameters to further optimize the local model and improve the accuracy of the local model. until the accuracy of the local model of the data source meets the requirements, the local model parameters can be changed. stored in data blocks 2 to ; the global model updated in each period is encrypted with a private key and uploaded to the corresponding block. its nodes on the blockchain use a consensus mechanism based on the quality of training parameters for verification. If  nodes consider that the updated global model parameters in this period are compared with the updated global model in the previous period, and the accuracy fluctuates within an acceptable range, then the global model parameters updated in this period can be compared and stored in the data block n corresponding to the block, see the blue part in Fig 1.

  1. Section 2.1 and Figure 1 need  revision. Though all the steps are mentioned but it is difficult to read and follow the flow chat in the Figure1. Number the subsections and each of the subsection describe model transfer phase, model training phase and model storage stage.

For this opinion, the details in Figure 1 have been modified, and the details have been modified in Section 2.1. Figure 1 is modified as follows.

Fig.1 Blockchain based differential optimization joint incremental learning algorithm framework

At the same time, the titles of three stages are added, namely model transmission stage, model training stage and model storage stage.